# Malic enzyme 1 knockout has no deleterious phenotype and is favored in the male germline under standard laboratory conditions

Jonathan M. Alektiar[1]*, Mengrou Shan[1]*, Megan D. Radyk[1], Li Zhang[1], Christopher J. Halbrook[1], Lin Lin[1], Carlos Espinoza[2], Ivan F. Mier[3], Brooke L. Lavoie[1], Lucie Salvatore[1], Marina Pasca di Magliano[2,4], Lewis C. Cantley[5,6], Jacob L. Mueller[3], Costas A. Lyssiotis[1,4,7]*

1 Department of Molecular & Integrative Physiology, University of Michigan, Ann Arbor, Michigan, United States of America, 2 Department of Surgery, University of Michigan, Ann Arbor, Michigan, United States of America, 3 Department of Human Genetics, University of Michigan, Ann Arbor, Michigan, United States of America, 4 Rogel Cancer Center, University of Michigan, Ann Arbor, Michigan, United States of America, 5 Department of Cancer Biology, Dana Farber Cancer Center, Boston, Massachusetts, United States of America, 6 Department of Cell Biology, Harvard Medical School, Boston, Massachusetts, United States of America, 7 Division of Gastroenterology, Department of Internal Medicine, University of Michigan, Ann Arbor, Michigan, United States of America

☉ These authors contributed equally to this work.
* clyssiot@med.umich.edu

**Data Availability Statement:** All relevant data are within the manuscript and its Supporting Information files.

## Abstract

Malic Enzyme 1 (ME1) plays an integral role in fatty acid synthesis and cellular energetics through its production of NADPH and pyruvate. As such, it has been identified as a gene of interest in obesity, type 2 diabetes, and an array of epithelial cancers, with most work being performed *in vitro*. The current standard model for ME1 loss *in vivo* is the spontaneous Mod-1 null allele, which produces a canonically inactive form of ME1. Herein, we describe two new genetically engineered mouse models exhibiting ME1 loss at dynamic timepoints. Using murine embryonic stem cells and Flp/FRT and Cre/loxP class switch recombination, we established a germline *Me1* knockout model (Me1 KO) and an inducible conditional knockout model (Me1 cKO), activated upon tamoxifen treatment in adulthood. Collectively, neither the Me1 KO nor Me1 cKO models exhibited deleterious phenotype under standard laboratory conditions. Knockout of ME1 was validated by immunohistochemistry and genotype confirmed by PCR. Transmission patterns favor *Me1* loss in Me1 KO mice when maternally transmitted to male progeny. Hematological examination of these models through complete blood count and serum chemistry panels revealed no discrepancy with their wild-type counterparts. Orthotopic pancreatic tumors in Me1 cKO mice grow similarly to Me1 expressing mice. Similarly, no behavioral phenotype was observed in Me1 cKO mice when aged for 52 weeks. Histological analysis of several tissues revealed no pathological phenotype. These models provide a more modern approach to ME1 knockout *in vivo* while opening the door for further study into the role of ME1 loss under more biologically relevant, stressful conditions.

**Funding:** M.S. was supported by an F32 fellowship from the NIH/NCI (5F32CA247492). M.D.R. receives funding from the National Institute of Child Health and Human Development (Training Program in Organogenesis, T32HD007505) and the National Cancer Institute (F32CA275283). C.A.L and L.C.C. were supported by a Mark Foundation ASPIRE award and a research partnership alliance with Astellas Pharmaceuticals. Research reported in this publication was supported by the University of Michigan Transgenic Animal Model Core and the Biomedical Research Core Facilities and by the National Cancer Institute of the National Institutes of Health under award number P30CA046592. PhenoMENA, non-invasive blood pressure, and rotarod testing was performed by the Physiology Phenotyping Core at the University of Michigan Medical School, which is supported in part by the Michigan Musculoskeletal Health Center (NIH P30 AR069620). The funders had no role in study design, data collection and analysis, or the content and publication of this manuscript.

**Competing interests:** L.C.C. and C.A.L. are inventors on patents pertaining to Kras regulated metabolic pathways, redox control pathways in pancreatic cancer, and targeting GOT1 or ME1 as a therapeutic approach (US Patent No: 2015126580-A1, 05/07/2015; US Patent No: 20190136238, 05/09/2019; International Patent No: WO2013177426-A2, 04/23/2015). L.C.C. owns equity in, receives compensation from, and serves on the Scientific Advisory Boards of Faeth Therapeutics, Agios Pharmaceuticals, Volastra Therapeutics, and Larkspur Biosciences. L.C.C.'s laboratory has previously received financial support from Petra Pharmaceuticals. Agios Pharmaceuticals is identifying metabolic pathways of cancer cells and developing drugs to inhibit such enzymes to disrupt tumor cell growth and survival. In the last 3 years, C.A.L. has received consulting fees from Astellas Pharmaceuticals, Odyssey Therapeutics, Third Rock Ventures, and T-Knife Therapeutics. This does not alter our adherence to PLOS ONE policies on sharing data and materials.

# Introduction

Malic enzyme 1 (ME1), or NADP+ dependent malic enzyme, is a cytosolic enzyme that plays a central role in cellular metabolism. ME1 is the cytosolic form of malic enzyme, with ME2 and ME3 localized to the mitochondria [1]. Responsible for the oxidative decarboxylation of malate to pyruvate, ME1 concurrently reduces NADP+, making it a valuable source of cytosolic NADPH and therefore, a highly conserved gene across mammalian species [1]. Indeed, isoforms of cytosolic malic enzyme are preserved in genomes back to single-celled yeasts [2].

ME1 activity impacts cellular metabolism both via its role in NADPH generation and its production of pyruvate. Knockdown of ME1 has been shown to lower cytosolic NADPH levels, decreasing the availability of this important metabolite in cellular redox homeostasis [3–5]. Further, the NADPH produced by ME1 has been proposed to directly feed into fatty acid biosynthesis, specifically in the synthesis of long-chain saturated fatty acids, though the mechanism for selectivity remains unclear [6]. The production of pyruvate from ME1 furthers its metabolic impact on the cell, as the pyruvate produced is shuttled into the mitochondria for production of acetyl-CoA, linking cytosolic and mitochondrial metabolism [7, 8].

The reprogramming of energy metabolism is hallmark of cancer [9–11]. Given both the position of ME1 in central carbon metabolism, and the recognition that it is upregulated in epithelial cancers, it has received considerable interest as a therapeutic target in oncology [7, 12–14]. Accordingly, drug development efforts have aimed to inhibit ME1 as a form of targeted therapy [14–16]. While targeting metabolic enzymes for cancer treatment has the potential to exploit metabolic pathways exclusive to the cancer cell, these proteins are still active in normal cells, which highlights the potential for untoward toxicities [17–19]. As such, an analysis of *Me1* knockout in mice was employed to ascertain if any such toxicities exist and, therefore, the suitability of ME1 as a drug target.

A spontaneous *Me1* knockout mouse model was reported in 1980, and this has previously been used to study the role of ME1 in fatty acid biosynthesis and links to obesity [20–22]. The knockout model involved the Mod-1 null mutation, a spontaneously duplicated sequence within the *Me1* gene, resulting in an inactive, larger malic enzyme being present in the cytosol [23]. Homozygous Mod-1 mice significantly lost overall body weight on a medium high fat diet [24]. Mod-1 mutation was also found to change adiposity in conjunction with a high-fat, non-soy protein diet, particularly in the retroperitoneal fat pad [22]. Under normal conditions, however, no further phenotype has been observed, and the Mod-1 mice age to adulthood and are reproductively competent [20].

Transmission ratios in Mod-1 null mice were noted to be generally consistent, and ME1 loss did not negatively affect fertility [20]. In some cases, gene mutation or loss can in fact improve fertility. This type of phenomenon, in which a mutant allele skews transmission away from the expected inheritance pattern is designated to be a transmission ratio distortion (TRD) [25]. TRDs are rare in mammals, with the best studied example being the *t*-complex [26–29]. In this example, a mutant allele called the *t*-haplotype on mouse chromosome 17 promotes flagellar motility in mutant sperm cells, enhancing their transmission to up to 99% of their progeny [26]. While not observed in Mod-1 null mice, loss of functional ME1 did not seem to be a detriment in a laboratory setting.

New approaches in genetic engineering allow for precise modification of the *Me1* gene in mice [30]. The Cre/loxP and Flp/FRT genetic engineering techniques—derived from P1 bacteriophages and *Saccharomyces cerevisiae*, respectively—provide dynamic approaches for site- and tissue-specific modification or deletion of genes via immunoglobin class switch recombination [31–33]. When combined, the orthogonal Cre/loxP and Flp/FRT recombination systems provide additional capabilities.

For example, a FRT-flanked gene trapping cassette followed by a loxP flanked (floxed) target exon creates a bifold approach for genetic knockout. In its basic state, the gene trapping cassette disrupts transcription of the target exon while expressing a reporter within the cassette. This modification results in a germline knockout of the target gene, removing it from the organism in its embryonic state. However, the FRT-flanked cassette can be removed though Flp/FRT recombination, in which a flippase-expressing mouse cell will remove the cassette in the germline. In this modified state, the expression of the critical exon is dependent on Cre/loxP recombination, which operates under similar principles—floxed genes are removed in the presence of Cre recombinase. An additional layer of complexity can be added to the system through the tamoxifen-inducible Cre-ERT2 transgene [34]. Using this model, genes can be knocked out in a time-dependent manner based on the timing of tamoxifen administration. Such an approach illustrates how adult knockout of the target gene would affect the animal's physiology, akin to pharmacological inhibition of the target protein.

Here, we outline two new mouse models accounting for *Me1* knockout in the germline and induced knockout as adults. We examined a panel of behavioral, histological, and biochemical features in the knockout mice under normal, unperturbed housing conditions on an ad libitum standard chow diet. In one model, we find that ME1 loss is not only unharmful to fertility as observed in Mod-1 null mice but is rather selected for in the male germline. With both models, we did not observe any consistent, significant detriments on physiological parameters or functions. This paper serves to document the exploration of possible phenotypes of Me1 deficient mice to better ascertain the role of the enzyme in mammals and establish any impacts of its loss in vivo.

## Materials and methods

### Mice

Mouse experiments were conducted under the guidelines of the Office of Laboratory Animal Welfare and approved by the Institutional Animal Care and Use Committees (IACUC) at the University of Michigan under protocol PRO00010606. All mice were of C57BL/6 background and were crossed to a full BL6J background using C57BL/6J mice, kept in ventilated racks in a pathogen-free animal facility with a 12 h light/12 h dark cycle, 30–70% humidity, and 68–74°F temperatures maintained in the animal facility at Rogel Cancer, University of Michigan. Mice were overseen by the Unit for Laboratory Animal Medicine. Cages were supplied with soft bedding (ScottPharma, Pure-o'Cel PB) and nesting material (WF Fisher, Enviropak) to benefit animal welfare. Single-housed animals were provided with additional enrichment in the form of a mouse igloo (Bio-Serv, K3327) or nestlet (Ancare, NES3600).

When necessary, mice were sacrificed via isoflurane jar or carbon dioxide chamber methods. For the former method, a 500 mL jar with 300 μL of isoflurane was kept in an upward flowing BSL-2 hood. Mice euthanized by this method were placed in the jar one at a time with the lid tightly sealed and separation of the mouse from the isoflurane pad itself. For the carbon dioxide chamber, mice were euthanized in their home cages at a flow rate of 2.5 L/min with no more than five mice at a time for at least ten minutes. Rear foot pinch was performed to ensure no sensation was present. Secondary euthanasia was performed by cervical dislocation or bilateral pneumothorax.

### Malic enzyme 1 mutant alleles

Malic enzyme 1 mutant mice were generated by the Transgenic Animal Model Core at the University of Michigan using genetically engineered ES cells from EUCOMM (HEPD0722) [35]. ES cells were placed in culture as described [36]. ES cell clones were microinjected into

albino C57BL/6 blastocysts to produce mouse ES cell-chimeras, as described [37]. ES cell-mouse chimeras were mated with FlpO transgenic mice [38] that had been backcrossed onto an albino C57BL/6 background so that germline transmission and elimination of the drug selection cassette occurred simultaneously to produce mice carrying the floxed *Me1*<Tm1c> allele on a C57BL/6 background.

Homozygosity of the *Me1*<tm1a> germline knockout allele (*Me1$^-$*; Me1 null) creates a whole-body knockout mouse model (Me1 KO). The *Me1*<Tm1c> floxed allele (*Me1$^{flox}$*) exhibits two loxP sites on the same strand, enabling gene excision by Cre recombinase.

## Me1 conditional knockout mouse model

The conditional global knockout model utilized transgenic mice with a tamoxifen inducible Cre-ERT2 controlled by the human ubiquitin C (UBC) promoter [34]. UBC-Cre-ERT2 mice were bred with Me1 floxed allele, generating the UBC-Cre-ERT2; *Me1*$^{flox/flox}$ mouse model, referred to as Me1 cKO. UbCre/Me1 flox mice were aged to 8–12 weeks, then were administered 5 mg tamoxifen by oral gavage daily for five days. Animals were then fed 400 mg tamoxifen citrate per kg chow for 3 weeks to achieve whole-body Cre recombination for *Me1* knockout.

## Genotyping

In-house PCR was run using DNA extracted from tail tissue. Thermal cycling was performed using the *Power* SYBR Green 2X Master Mix (ThermoFisher) according to the provided protocol on a SimpliAmp Thermal Cycler (ThermoFisher). Gel electrophoresis was performed on 2.5% agarose gel with 10,000X SYBR Safe DNA gel stain (Invitrogen). 1 kb and 100 bp DNA Ladders (New England Biolabs) were used to measure amplicon size. Gels were run at 125V for 75 minutes and imaged using the ChemiDoc Imaging System (Bio-Rad). For routine cage management, genotyping was outsourced to a commercial vendor, who used real time PCR with specific probes designed for each gene (Transnetyx, Cordova, TN). The primers utilized for both in-house and outsourced PCR were as follows: Me1 WT and Me1 Flox (Fw-`TTTGTAGAAGGAAGCATCCCAG`, Rev-`CCTGGGCTTGAGATCCAACA`); Me1 Null (Fw-`CAACGGGTTCTTCTGTTAGTCC`, Rev-`CCTGGGCTTGAGATCCAACA`); Cre (Fw-`TCGCGATTATCTTCTATATCTTCAG`, Rev-`GCTCGACCAGTTTAGTTACCC`).

## Aging

Me1 KO mice (n = 8) and heterozygous *Me1$^{+/-}$* (Me1 Het) mice (n = 4) were aged for 18–23 months and were euthanized in accordance with colony consolidation measures in 2020. Additionally, Me1 cKO mice (n = 6) and *Me1$^{flox/flox}$* (Me1 FL/FL) mice (n = 7) were aged to 52 weeks and were assessed for a variety of physiological and behavioral phenotypes. Animals were not single housed when possible to benefit animal welfare. Animals were monitored by ULAM veterinary staff daily for significant behavioral changes or signs of suffering as dictated by the ULAM End-Stage Illness Scoring System, assessing degree of morbidity based on appearance, physical exam parameters, natural and provoked behaviors, and body condition score. Mice are euthanized within 24 hours of discovery of such changes in welfare. No mice in either of these studies met the criteria for humane endpoints and early euthanization, nor were any found dead by veterinary staff.

## Histology

Me1 KO mouse tissues were harvested and fixed in zinc formalin fixative solution (Z-FIX, Cancer Diagnostics) for 24 hours. Fixed tissues were then transferred to 70% ethanol and

embedded in paraffin. The blocks were processed using a Leica ASP300S tissue processor (Leica Microsystems) and sliced at 4 μm. H&E staining was performed using Mayer's hematoxylin solution and Eosin Y (ThermoFisher).

For the Me1 cKO mice, necropsy, histology, and pathology evaluation was performed by the In Vivo Animal Core (IVAC) laboratory within the Unit for Laboratory Animal Medicine at Michigan Medicine. Animals were humanely euthanized in an isoflurane drop jar. At gross necropsy, tissues were collected, and immersion fixed in 10% neutral buffered formalin (NBF) for 24hrs. Following fixation, bone (sternum) was decalcified in Immunocal for overnight. Formalin-fixed tissues were processed through graded alcohols and cleared with xylene followed by infiltration with molten paraffin using an automated VIP5 or VIP6 tissue processor (TissueTek, Sakura-Americas, Torrance, CA). Following paraffin embedding using a Histostar Embedding Station (ThermoScientific, Hanover Park, IL), tissues were sectioned at 4 μm on an HM-355S rotary microtome (ThermoFisher Scientific, Hanover Park, IL). Following deparaffinization and hydration with xylene and graded alcohols, formalin-fixed, paraffin embedded (FFPE) slides were stained with Harris hematoxylin (ThermoFisher Scientific, Cat# 842), differentiated with Clarifier (ThermoScientific, Cat#7401), blued with bluing reagent (ThermoFisher Scientific, Cat#7301), stained with eosin Y, alcoholic (ThermoFisher Scientific, Cat# 832), then dehydrated and cleared through graded alcohols and xylene and coverslipped with Micromount (Leica cat# 3801731, Buffalo Grove, IL) using a Leica CV5030 automatic coverslipper. A board-certified veterinary anatomic pathologist evaluated the slides.

All slides were stained for ME1 using the Discovery Ultra XT autostainer (Ventana Medical Systems). IHC was performed for ME1 (1:500 and 1:250; Abcam, ab97445) and counterstained with Mayer's hematoxylin (Sigma). IHC slides were then scanned on a Pannoramic SCAN scanner (Perkin Elmer) or imaged on a BX53 Light Microscope using cellSens Imaging Software (Olympus Life Science).

## NADP(H) measurements

NADPH and NADP+ abundances were measured in male and female wildtype and Me1-/- animals aged 12–14 weeks. Immediately following isoflurane overdose, organs were collected, cut into approximately equal sizes, weighed, and flash frozen in liquid nitrogen. Metabolite extraction from the full cohort was performed on the same day. Frozen organs were transferred to 2mL round bottom tubes with locking caps containing a 5mm stainless steel metallic bead. Dry-ice cold 80% methanol (in water) was added to each tube and samples were shaken and homogenized on a TissueLyser II (Qiagen) in intervals of 30 seconds, with rests on dry ice, until fully homogenized. Samples were centrifuged at maximum speed for 10 minutes at 4°C and supernatant was collected. The volume of supernatant to collect for drying was normalized within organ groups and based on tissue weight prior to extraction. Normalized supernatant volumes were dried with a Savant SpeedVac Integrated Vacuum Concentrator (model SPD1030). Dried pellets were reconstituted with 50% v/v methanol in water for LC-MS.

NADPH and NADP+ were analyzed by liquid chromatography—mass spectrometry (LC-MS). For LC, a Waters Xbridge BEH Amide 2.5 μm 2.1 X 150 mm Column XP and a Phenomenex High Pressure column protection filter were used for the separation. Solvent A consisted of Water with 20 mM ammonia acetate (LCMS grade, Sigma) and 20 mM ammonia (pH 9.5; LCMS grade, Millipore). Solvent B was pure acetonitrile (LCMS grade, Supelco). Pump Seal wash and autosampler wash were done with 50% isopropanol (Fisher) and 0.1% formic acid (LCMS grade, Fisher). The LC profile was performed at 0.15 ml/min: for 0–1 min, solvent was 90% B; at 5 min, 50% B; at 10min, 5% B; at 12 min, 5% B; at 12.1 min, 90% B; at 15

**Table 1. LC-MS detection parameters for NADP+ and NADPH.**

| Compound | Retention Times (min) | Retention Time Window (min) | Precursor (m/z) | Product (m/z) | Collision Energy (V) |
|---|---|---|---|---|---|
| NADP+_Pos | 8.1 | 3 | 744.083 | 507.883 | 29.72 |
| NADP+_Pos | 8.1 | 3 | 744.083 | 603.967 | 20.27 |
| NADPH_Pos | 7.9 | 3 | 746.098 | 301.967 | 36.69 |
| NADPH_Pos | 7.9 | 3 | 746.098 | 728.967 | 18.75 |

m/z, mass/charge; V, voltage.

min, 90% B. The column compartment temperature was maintained at 30˚C and the autosampler at 5˚C. Injection volume was 3 μL.

For MS, a TSQ Quantis Triple Quad MS was employed. It was calibrated with Pierce Triple Quadrupole Calibration Solution External Mass Range with a microESI probe from Thermo Scientific. Infusion rate was 3 μL/min. Source parameters were as follows: H-ESI Positive Static Voltage 3500 V; Sheath Gas (Arb) 40, Aux Gas 5, Sweep Gas 1; Ion transfer Tube Temp 325˚C and Vaporizer Temp 350˚C. Dwell Time was 80 ms. Q1 and Q3 FWHM were at 0.7 da. CID gas (mTorr) was at 1.5 with helium.

LCMS grade methanol was from Sigma. The water was from a Millipore purified filter device with 18 omegas. NADPH and NADP+ are from Cayman with highest purity. Detection details are in Table 1. Thermo Scientific Triple Quad TSQ Quantis LC/MS/MS system consists of a Vanquish binary pump, a Vanquish autosampler, and a Vanquish column compartment with a switch valve for two column setup configurations. Thermo Scientific Excalibur Version 4.4.16.14 data system with Dell computer Opex-EX3 with Win10 operation system was used for calibration, compound optimization and sample data acquisition. Data processing was performed by Thermo Scientific Tracing finder 3.1 and Skyline 12.1.

## Hematological tests

For complete blood counts (CBCs): whole blood was collected into K3 EDTA anticoagulant tubes (Microvette 20.1288.100) and CBCs were run on a Heska Element HT5 (Heska Corporation, Loveland, CO) automated veterinary hematology analyzer. For serum chemistry tests: whole blood was collected into serum separator tubes (BD Microtainer 365967), allowed to clot, and separated into serum by centrifugation. Serum chemistries were run on an Liasys 330 (AMS Alliance, Guidonia, Italy) automated wet chemistry analyzer. Assays were performed within the ULAM In Vivo Animal Core pathology laboratory at the University of Michigan. Quality control was performed daily using manufacturer-provided reagents and the laboratory is a participant in an external independent quarterly quality assurance program (Veterinary Laboratory Association Quality Assurance Program).

## Behavioral and clinical analyses

Me1 cKO and controls were aged to 52 weeks (knockout or mock initiated at 8–12 weeks of age, see relevant section above), at which point they were screened for several functional measures. This included a modified SHIRPA screen using the PhenoMENA Clinical Examination and Behavioral Assessment, non-invasive blood pressure (NIBP) measurements, and the rotarod performance test [39–41]. All analyses were performed by the Physiology Phenotyping Core at the University of Michigan Medical School.

The PhenoMENA assessment is composed of a variety of tests. Observation of baseline bodily function and activity abnormalities in a small viewing jar (40 x 20 cm) for five minutes

and a large grid-like arena (55 x 33 x 18 cm). After 10 seconds in the arena test, activity levels (GRiD) are quantified by how many squares of the arena are covered by the mouse in 30 seconds. Tail suspension was used to screen for proper stress and arousal responses, as well as a grip strength test, repeated in triplicate. Supine restraint was used to monitor smaller physical abnormalities and reflexes. Each behavior was scored and categorized as good behaviors (GS), moderate behaviors (MS), and bad behaviors (BS). Mouse grip strength (GRiP) was reported in kilograms.

Non-invasive blood pressure (NIBP) measurements, including mean systolic and diastolic blood pressures, mean arterial pressures, and heart rate, were taken using CODA NIBP System. The rotarod performance test assessed the ability to stay on a constantly accelerating rotating cylindrical platform and their acclimation to the test over time. Performance was reported as the longest number of seconds each mouse stayed on the rotarod to account for test acclimation.

## Orthotopic injection

Mice were anesthetized with ventilator induction of isoflurane and carprofen as analgesic. An incision was made in the peritoneal cavity. The spleen was eviscerated along with a small portion of the pancreatic tail. 10,000 $Kras^{G12D/+}$; $Trp53^{R172H/+}$; $Ptf1a^{Cre-ERTM}$ cells were suspended in 25 µL of PBS and Matrigel before injection in the distal pancreas. Mice were monitored regularly and taken down four weeks after the operation.

## Data analysis and presentation

Figs 1A–1C and 2A, 2C and 2D were created with BioRender.com. All other figures and statistics were created in GraphPad Prism. Unless otherwise noted, error bars and listed error margins represent standard error of the mean. All $p$ values were determined using multiple t-tests not assuming a consistent standard deviation except for the TRD data, which was generated using chi square analysis. False discovery rate (FDR) was accounted for with discovery and $q$ values determined using the Two-stage linear step-up procedure of Benjamini, Krieger and Yekutieli, with Q = 1% [42].

## Results

### Generation of Me1 knockout models

Embryonic stem cells genetically engineered to model ME1 loss were purchased from EUCOMM [43, 44]. The cells were injected into murine blastocysts to generate chimeric mice containing an *Me1* gene flanked by loxP sites. This was preceded by a gene trapping cassette flanked by flippase recognition target (FRT) sites. The chimeric mice were mated with FlpO-10 mice to generate two strains of mice.

Two *Me1* knockout alleles were generated by this crossing: the *Me1* germline knockout allele (Me1 null) and the *Me1* floxed allele (Me1 flox) (**Fig 1A**). The Me1 null allele has the L1L2_Bact_P cassette, containing a lacZ reporter and neomycin resistance (Neo) gene flanked by FRT sites. The cassette was inserted at location 86655157 prior to the critical exon 4 in Me1. The presence of this cassette prevents proper transcription of the exon and precludes the presence of ME1. Homozygosity for this allele generates the *Me1* germline knockout model (Me1 KO) (**Fig 1B**). In the *Me1* floxed allele, the nonsense mutation was removed by FlpO-FRT recombination, leaving a functional, floxed *Me1* gene. Crossing in a whole-body, tamoxifen inducible Cre recombinase allele, UBC-Cre-ERT2, generates the UBC-Cre *Me1* conditional

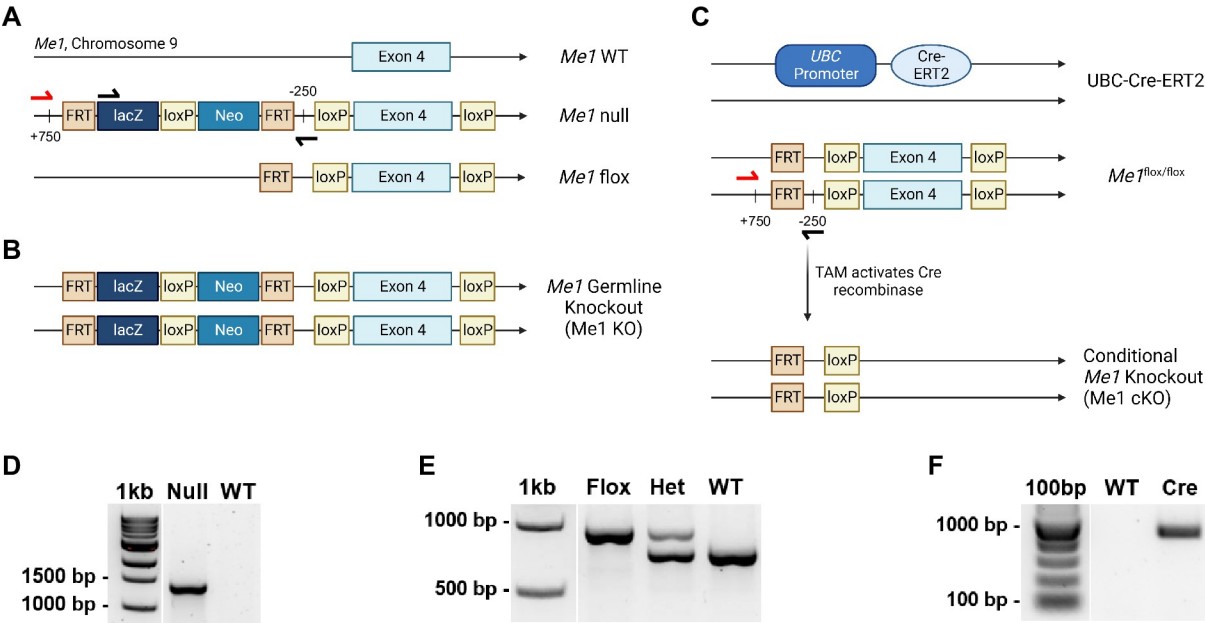

**Fig 1. *Me1* knockout models.** (A) Representation of the wild-type *Me1* gene (top) and two *Me1* knockout alleles, Me1 null (middle) and Me1 flox (bottom). (B) The *Me1* germline knockout model (Me1 KO). (C) The UBC-Cre *Me1* conditional knockout model (Me1 cKO). Arrows represent primer locations; presentation is not to scale. (D-F) Genotype validation of (D) the Me1 null lacZ reporter, (E) Me1 flox and wild-type, and (F) UBC-Cre alleles by PCR of tail DNA. The black forward primer and the black reverse primer produce a ~1.5Kbp band for the null allele. The red forward primer and the black reverse primer produce a ~1Kbp band for the floxed allele and a ~750bp band for the wild type allele. The UBC-CreERT2 allele is transgenic.

knockout model (Me1 cKO) (**Fig 1C**). All alleles were validated by PCR (**Fig 1D–1F**). The two approaches to *Me1* knockout provide flexibility in how ME1 deletion can be examined.

## Me1 germline knockout model

The *Me1* germline knockout model (Me1 KO) utilizes the FRT-flanked gene trapping cassette prior to exon 4 to knock out the Me1. To determine if the focused deletion of *Me1* was compatible with life, as has been reported using the spontaneous *Me1* mutation model [23], we crossed mice heterozygous for the Me1 KO allele. From this analysis, we observed that Me1 KO mice were born and age to adulthood without observable phenotype under normal conditions (**S1 Table**).

Indeed, complete blood count values were consistent between Me1 KO mice and wild-type C57BL/6J animals (**Table 2**). Similarly, when we crossed Me1 KO mice, we observed that the resultant litters were of typical size for C57BL/6J mice, and the pups mature to adulthood and are grossly normal (**S1 Fig**). This indicated that Me1 KO did not negatively impact fecundity.

Strikingly, however, when performing dihybrid crosses for the Me1 null allele (**Fig 2A**), non-Mendelian transmission patterns were observed in the progeny that favored the Me1 null allele. In a group of 142 pups born from heterozygous Me1 KO (Het) mice, there was a preference towards transmitting the Me1 null allele (**Fig 2B**). This was borderline significant in the total population ($p = 0.052$) and reached statistical significance when separated by sex, with a preference for male *Me1* KO offspring ($p = 0.0187$).

Pursuing the possibility of a transmission ratio distortion (TRD), we followed the example of the *t*-complex TRD, in which paternal transmission of the mutant *t*-haplotype improved sperm cell flagellar motility, outcompeting cells without the mutation. To that end, we crossed

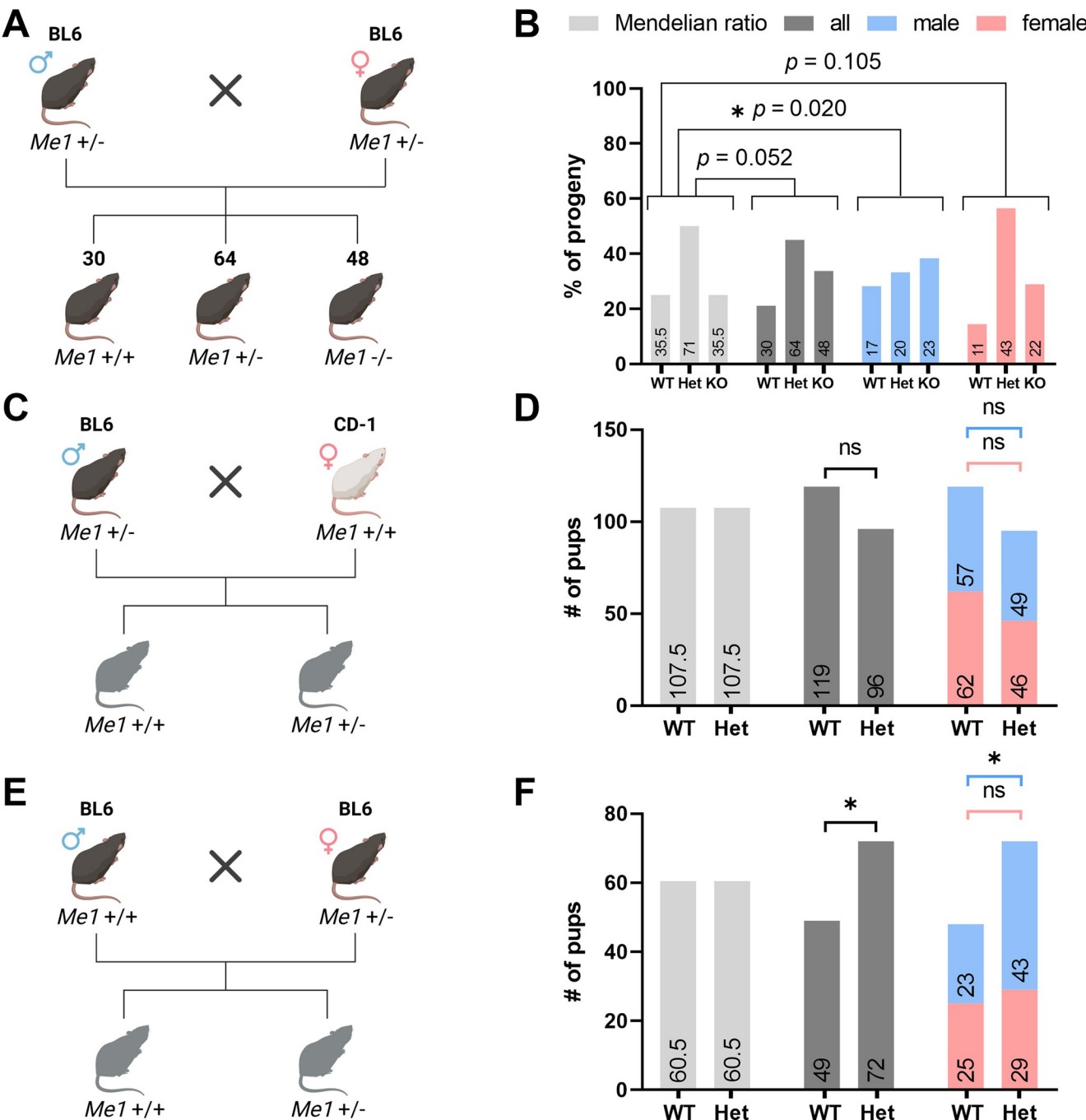

**Fig 2. Me1 KO transmission patterns.** (A) Dihybrid cross breeding scheme for mice containing one Me1 null allele and one wild-type allele. (B) Percentage of pups (n = 142) and their corresponding genotypes compared to expected Mendelian ratios and broken down by sex. Non-Mendelian transmission of the Me1 null allele is noted, with a greater than expected number of Me1 KO and heterozygous Me1 KO (Het) mice in male progeny. (C) Breeding scheme for paternal inheritance of the Me1 null allele. (D) Number of pups (n = 215) and corresponding genotypes compared to Mendelian ratios. No significant trends noted. (E) Breeding scheme for maternal inheritance of the Me1 null allele. (F) Number of pups (n = 121) and corresponding genotypes compared to Mendelian ratios. Non-Mendelian transmission noted in pups overall (*p* = 0.0451), particularly in male progeny (*p* = 0.0187).

an Me1 Het male with a CD-1 female wild-type mouse to produce larger litters and control for background-specific phenotype (**Fig 2C**). No TRD was observed when the Me1 null allele was paternally transmitted and was similarly not observed when broken down by sex of progeny

**Table 2. Complete blood count values from Me1 WT and Me1 KO mice at 8 weeks of age.**

| CBC Value | WT (n = 5) | KO (n = 8) | *p* value | *q* value | Discovery |
|---|---|---|---|---|---|
| WBC (K/µL) | 6 ± 1 | 11 ± 1 | 0.04 | 0.15 | No |
| NEU (K/µL) | 0.7 ± 0.1 | 2.6 ± 0.7 | 0.06 | 0.15 | No |
| LYM (K/µL) | 5 ± 1 | 8 ± 2 | 0.23 | 0.42 | No |
| MONO (K/µL) | 0.13 ± 0.01 | 0.24 ± 0.03 | 0.02 | 0.14 | No |
| EOS (K/µL) | 0.036 ± 0.005 | 0.09 ± 0.02 | 0.06 | 0.15 | No |
| BAS (K/µL) | 0.006 ± 0.002 | 0.04 ± 0.01 | 0.02 | 0.14 | No |
| RBC (M/µL) | 9.13 ± 0.08 | 9.2 ± 0.1 | 0.55 | 0.70 | No |
| HGB (g/dL) | 14.6 ± 0.3 | 15 ± 0.2 | 0.25 | 0.42 | No |
| HCT (%) | 41.2 ± 0.5 | 41.6 ± 0.5 | 0.63 | 0.74 | No |
| MCV (fL) | 45.2 ± 0.5 | 45.1 ± 0.2 | 0.81 | 0.82 | No |
| MCH (pg) | 16 ± 0.2 | 16.25 ± 0.08 | 0.24 | 0.42 | No |
| MCHC (g/dL) | 35.4 ± 0.3 | 36 ± 0.1 | 0.03 | 0.15 | No |
| RDW % (%) | 13.1 ± 0.2 | 13.4 ± 0.3 | 0.33 | 0.49 | No |
| PLT (K/µL) | 1100 ± 100 | 1020 ± 80 | 0.78 | 0.82 | No |
| MPV (fL) | 5.06 ± 0.05 | 5.13 ± 0.05 | 0.37 | 0.52 | No |

Discovery determined using Two-stage linear step-up procedure. Q = 1%. WBC, white blood cells; NEU, neutrophils; LYM, lymphocytes; MONO, monocytes; EOS, eosinophils; BAS, basophils; RBC, red blood cell count; HGB, hemoglobin; HCT, hematocrit; MCV, mean corpuscular volume; MCH, mean corpuscular hemoglobin; MCHC, mean corpuscular hemoglobin concentration; RDW %, red blood cell distribution width; PLT, platelet count; MPV, mean platelet volume.

(**Fig 2D**). Checking to see if the TRD could be maternally influenced, we repeated the test cross in BL6J mice with the Me1 null allele coming from the female (**Fig 2E**). Here, a significant bias favoring the Me1 null allele was observed (**Fig 2F**). Like the initial dihybrid crosses, this phenomenon was almost exclusively observed in male progeny, confirming that ME1 loss is sometimes preferred in BL6J mice under laboratory conditions.

Analysis of a host of murine tissues for ME1 expression revealed the highest staining in liver, thymus and testes. Thus, we examined Me1 expression in these tissues by IHC (**Fig 3A–3C**). *Me1* genotype was tracked by PCR of tail tissue. Liver, thymus, and testes were harvested from Me1 KO, Me1 Het, and wild type mice. Organs were processed for into FFPE blocks and histology slides were generated and stained for H&E as well as IHC for ME1. *Me1* genotype correlated with ME1 expression, with Me1 Het tissues exhibiting partial expression and Me1 KO tissues showing total depletion of ME1. Additionally, we processed this suite of tissues, as well as kidney, pancreas and spleen, to analyze abundance of the ME1 cofactor NADP+ and NADPH by liquid chromatography-mass spectrometry (LC/MS). Across tissues, statistically significant differences in NADP+, NADPH, or the NADP+/NADPH ratios were not observed (**Fig 3D–3I** and **S2 Fig**). Lastly, a serum chemistry panel was run on blood obtained by cardiac puncture (**Fig 3J–3R**). Statistically significant hematological phenotypes were not observed across the array of metabolic markers.

## Me1 conditional knockout model

The embryonic knockout data presented provide support for the idea that ME1 inhibition may be a safe and therapeutically tractable drug target. However, a more accurate means to determine the therapeutic utility of ME1 inhibition is to delete expression in adult animals. Such an analysis provides a more accurate comparison to an adult taking a drug. This would preclude the possibility that during development, compensatory metabolic pathways were activated in the Me1 null animals that facilitated maturation to adulthood. To this end, we generated a

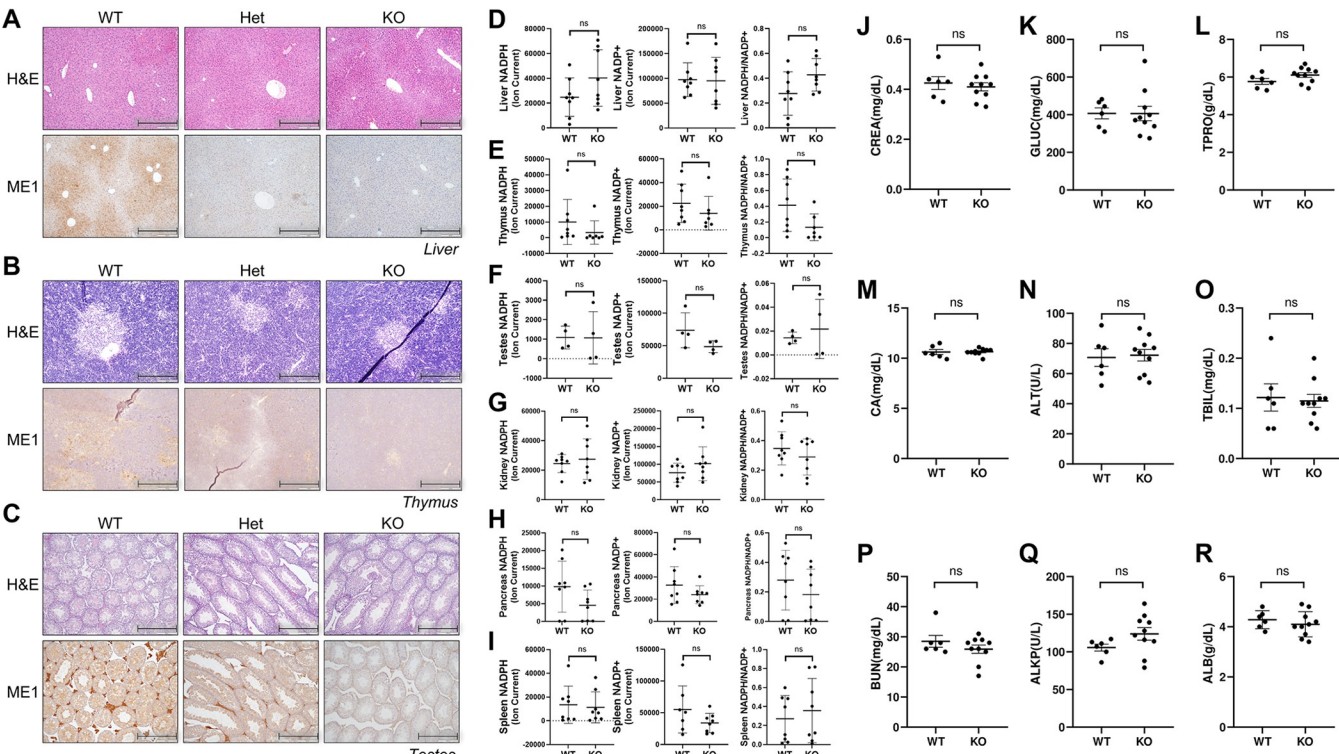

**Fig 3. ME1 protein expression is notably reduced in Me1 KO mice with no impact on the NADPH/NADP+ ratio.** (A-C) Me1 WT, Het, and KO mice were examined for histology and immunohistochemistry, and representative slides were taken. (A) The liver was imaged at 10X magnification (scale bar = 400 μm) with a 1:500 ME1 antibody stain ratio for IHC. (B) The thymus at 20X magnification (scale bar = 200 μm) with a 1:250 ME1 antibody stain ratio. (C) The testes at 10X (scale bar = 400 μm) with a 1:250 ME1 antibody stain ratio. Across the three tissues, Me1 Het mice exhibited partial knockout of ME1 and Me1 KO mice exhibited total knockout. NADP+, NADPH, and NADP+/NADPH ratios in (D) liver, (E) thymus, (F) testes, (G) kidney, (H) pancreas, and (I) spleen from Me1 wild type versus Me1 null animals, as assessed by LC-MS. Males and females were analyzed in equal proportion (S2 Fig) Serum chemistry assessment checked for (J) CREA, creatinine; (K) GLUC, glucose; (L) TPRO, total protein; (M) CA, calcium; (N) ALT, alanine transferase; (O) TBIL, bilirubin; (P) BUN, blood urea nitrogen; (Q) ALKP, alkaline phosphatase; and (R) ALB, albumin. Each data point represents an individual animal. Significance was determined by student's t-test.

cohort of *Me1* conditional knockout (Me1 cKO) mice (**Fig 1C**) (n = 6) and a cohort of *Me1*<sup>flox/flox</sup> (Me1 FL/FL) mice as normal controls (n = 7). When the animals reached 8–12 weeks of age, both groups were administered tamoxifen by oral gavage for 5 consecutive days. Then, mice were placed on a tamoxifen-diet for an additional 3 weeks.

The Me1 cKO mice were aged >1 year without observable phenotype, including no observed differences in ambulation, grooming, eating, and drinking. To explore the prospect of more subtle phenotypes, both groups were subject to a series of behavioral, movement-oriented, and clinical tests.

To test for behavioral phenotypes, Me1 cKO and Me1 FL/FL mice underwent a modified SHIRPA screen, starting with the PhenoMENA Clinical Examination and Behavioral Assessment [39–41]. Ultimately, Me1 cKO mice showed no statistically significant behavioral phenotype distinct from that of the normal control group (**Fig 4A**). Similar grip strengths (GRiP) and explorative proclivity (GRiD) scores were noted along with appropriate mean levels of good (GS), moderate (MS), and bad (BS) scoring behaviors in both groups. Further, the average behavioral scores across each test were within the normal range for BL6 mice.

Mice subsequently underwent the accelerating rotarod performance test to assess their balance and motor coordination. The best performance for each mouse was averaged within their genetic groups (**Fig 4B**). There was a slight difference between Me1 cKO and FL/FL groups

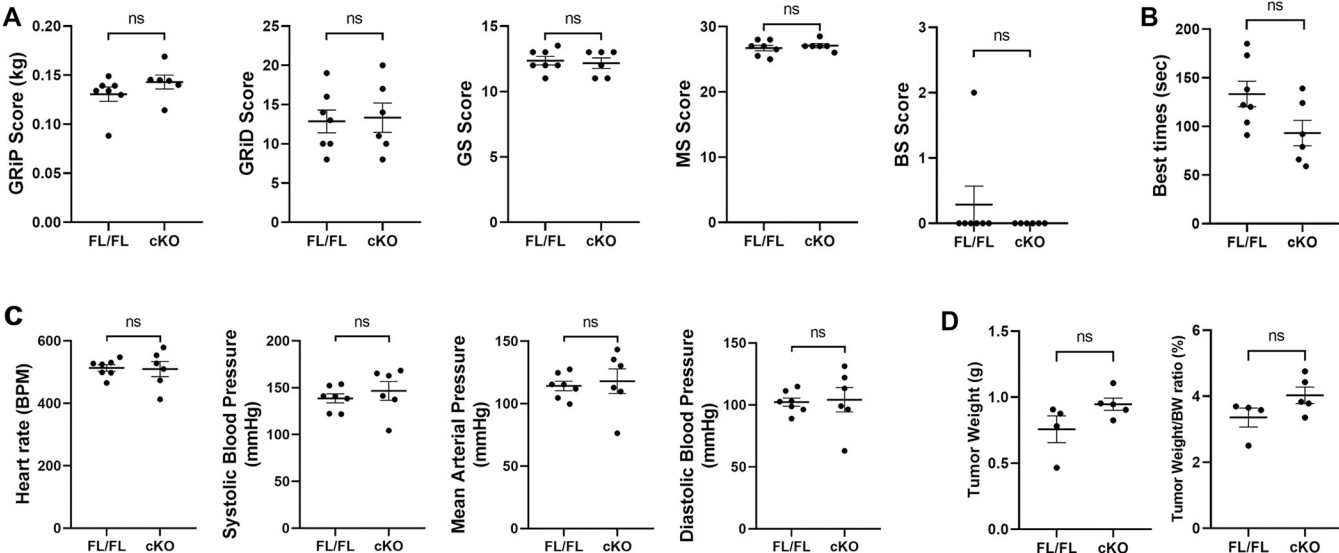

**Fig 4. Me1 cKO mice behavior and physiology is not statistically different from Me1 FL/FL mice.** (A) PhenoMENA behavioral analysis measured good scoring (GS), moderate scoring (MS), and bad scoring (BS) behaviors, along with grip strength (GRiP) and explorative proclivity (GRiD). GRiP scores: normal control (FL/FL) = 0.130 ± 0.007 kg; cKO = 0.143 ± 0.007 kg. No differences between groups noted, and all values were within normal ranges for BL6 mice. (B) Accelerating rotarod performance test. Best performances (sec) were recorded for each mouse and were averaged for both groups. FL/FL = 130 ± 10 sec; cKO = 90 ± 10 sec. (C) Average heart rate. FL/FL = 510 ± 10 BPM; cKO = 510 ± 20 BPM. Average systolic blood pressure. FL/FL = 139 ± 5 mmHg; cKO = 147 ± 10 mmHg. Average mean arterial pressure. FL/FL = 114 ± 4 mmHg; cKO = 118 ± 10 mmHg. Average diastolic blood pressure. FL/FL = 102 ± 3 mmHg; cKO = 104 ± 10 mmHg. (D) Murine KPC pancreatic tumor cells orthotopically implanted in mice were weighed and compared to body weight (BW). Average tumor weight. FL/FL = 0.8 ± 0.1 g; cKO = 0.94 ± 0.05. Tumor weight / body weight ratio. FL/FL = 3.4 ± 0.3%; cKO = 4.0 ± 0.2%.

that did not reach statistical significance ($p = 0.053$), with control mice averaging 130 ± 10 seconds and cKO mice averaging 90 ± 10 seconds for best performances. Both groups had noticeably large standard deviations ($SD_{FL/FL} = 35$ sec, $SD_{cKO} = 32$ sec).

We next assessed non-invasive blood pressure (NIBP) and cardiovascular performance. Statistically significant differences in mean systolic and diastolic blood pressures were not observed across groups (**Fig 4C**). Further, average mean arterial pressures and heart rates were not distinct. These factors confirm a strong circulatory similarity between the Me1 cKO and FL/FL mice.

A separate cohort of Me1 cKO (n = 5) and Me1 FL/FL (n = 4) mice aged to 14 weeks were injected with orthotopic pancreatic ductal adenocarcinoma cells to see how the tumors developed in the two groups. 10,000 $Kras^{G12D/+}$; $Trp53^{R172H/+}$; $Ptf1a^{Cre-ERTM}$ (KPC) cells in Matrigel were orthotopically injected into the distal pancreas. After one month, the tumors were excised and weighed with and without normalization for body weight (**Fig 4D**). In both measurements, there was no observed difference in tumor growth in the Me1 cKO mice.

After behavioral studies were completed on the first cohort, the Me1 cKO and Me1 FL/FL mice were sacrificed to examine blood, tissue, and organ histology and to quantify *Me1* knockout in *Me1*-expressing tissues. Similar to the Me1 KO mice, complete blood count values were not statistically different between genotypes (**Table 3**).

The liver, thymus, and testes were harvested and processed for hematoxylin and eosin (H&E) staining and immunohistochemistry with ME1 antibody staining (**Fig 5A–5C**). Malic enzyme 1 knockout was evident throughout the liver, thymus, and testes, as compared to the control animals. A serum chemistry panel was performed, and no hematological phenotype was observed (**Fig 5D–5L**). Further, detailed histological analysis by a board-certified

**Table 3. Complete blood count values from Me1 FL/FL and Me1 cKO mice.**

| CBC Value | FL/FL (n = 7) | cKO (n = 6) | p-value | q-value | Discovery |
|---|---|---|---|---|---|
| WBC (K/µL) | 6.8 ± 0.6 | 5 ± 1 | 0.17 | 0.59 | No |
| NEU (K/µL) | 1.1 ± 0.1 | 0.8 ± 0.3 | 0.35 | 0.75 | No |
| LYM (K/µL) | 4.9 ± 0.5 | 3.7 ± 0.8 | 0.23 | 0.59 | No |
| MONO (K/µL) | 0.5 ± 0.07 | 0.3 ± 0.08 | 0.09 | 0.59 | No |
| EOS (K/µL) | 0.2 ± 0.06 | 0.12 ± 0.02 | 0.23 | 0.59 | No |
| BAS (K/µL) | 0.039 ± 0.006 | 0.023 ± 0.007 | 0.13 | 0.59 | No |
| RBC (M/µL) | 9.3 ± 0.4 | 8.9 ± 0.8 | 0.70 | 0.93 | No |
| HGB (g/dL) | 14.4 ± 0.7 | 14 ± 1 | 0.64 | 0.93 | No |
| HCT (%) | 44 ± 2 | 42 ± 3 | 0.75 | 0.93 | No |
| MCV (fL) | 47 ± 0.6 | 47.6 ± 0.8 | 0.58 | 0.93 | No |
| MCH (pg) | 15.5 ± 0.2 | 15.5 ± 0.3 | 0.97 | 0.98 | No |
| MCHC (g/dL) | 33 ± 0.2 | 32.6 ± 0.3 | 0.14 | 0.59 | No |
| RDW% (%) | 15.8 ± 0.7 | 15.5 ± 0.7 | 0.80 | 0.93 | No |
| PLT (K/µL) | 1000 ± 200 | 1000 ± 300 | 0.97 | 0.98 | No |
| MPV (fL) | 5.6 ± 0.1 | 5.6 ± 0.2 | 0.77 | 0.93 | No |

Discovery determined using Two-stage linear step-up procedure. Q = 1%. WBC, white blood cells; NEU, neutrophils; LYM, lymphocytes; MONO, monocytes; EOS, eosinophils; BAS, basophils; RBC, red blood cell count; HGB, hemoglobin; HCT, hematocrit; MCV, mean corpuscular volume; MCH, mean corpuscular hemoglobin; MCHC, mean corpuscular hemoglobin concentration; RDW %, red blood cell distribution width; PLT, platelet count; MPV, mean platelet volume.

pathologist revealed no significant histological phenotype between the control and Me1 cKO mice in all examine organs at one year of age (S3 Fig).

## Discussion

In this study, we describe the development and characterization of two distinct *Me1* knockout mouse models to determine the impact of ME1 loss on organismal physiology and health. Using modern genetic engineering techniques, we confirmed a previous observation that embryonic *Me1* knockout is fully compatible with life [23]. Under routine housing and feeding conditions, the only observed phenotype was the transmission ratio distortion (TRD) in favor of the Me1 null allele. Selection for ME1 loss suggests that the role of malic enzyme 1 is either redundant or unnecessary when food intake is standard and no apparent threats to survival are present. In the adult, whole body, conditional knockout model, we again did not observe any deleterious phenotypes upon *Me1* loss in an analysis that included a cadre of behavioral, physiological, and histological assessments. Whether permissible by compensatory mechanisms of malic enzyme isoforms or simply a byproduct of the conditions studied, this study collectively demonstrates that ME1 may exhibit a remarkable safety profile as a putative drug target. Given its deep evolutionary conservation, future studies will be required to identify environmental stressors/context in which ME1 provides an organismal advantage.

### Knockout models

The current standard for *Me1* knockout in vivo is either the spontaneous Mod-1 mutant model [20–22] or in-situ hybridization for embryonic studies [45]. We provide new models to the research community for the study of Me1. The Me1 null, Me1 flox, and UBC-Cre-ERT2 alleles are transmitted easily in breeding and their genotypes are straightforward to identify by PCR. Both the Me1 KO and Me1 cKO models demonstrated thorough knockout in tissues expressing ME1 [46], including liver, thymus, and testes. The *Me1* germline knockout animals

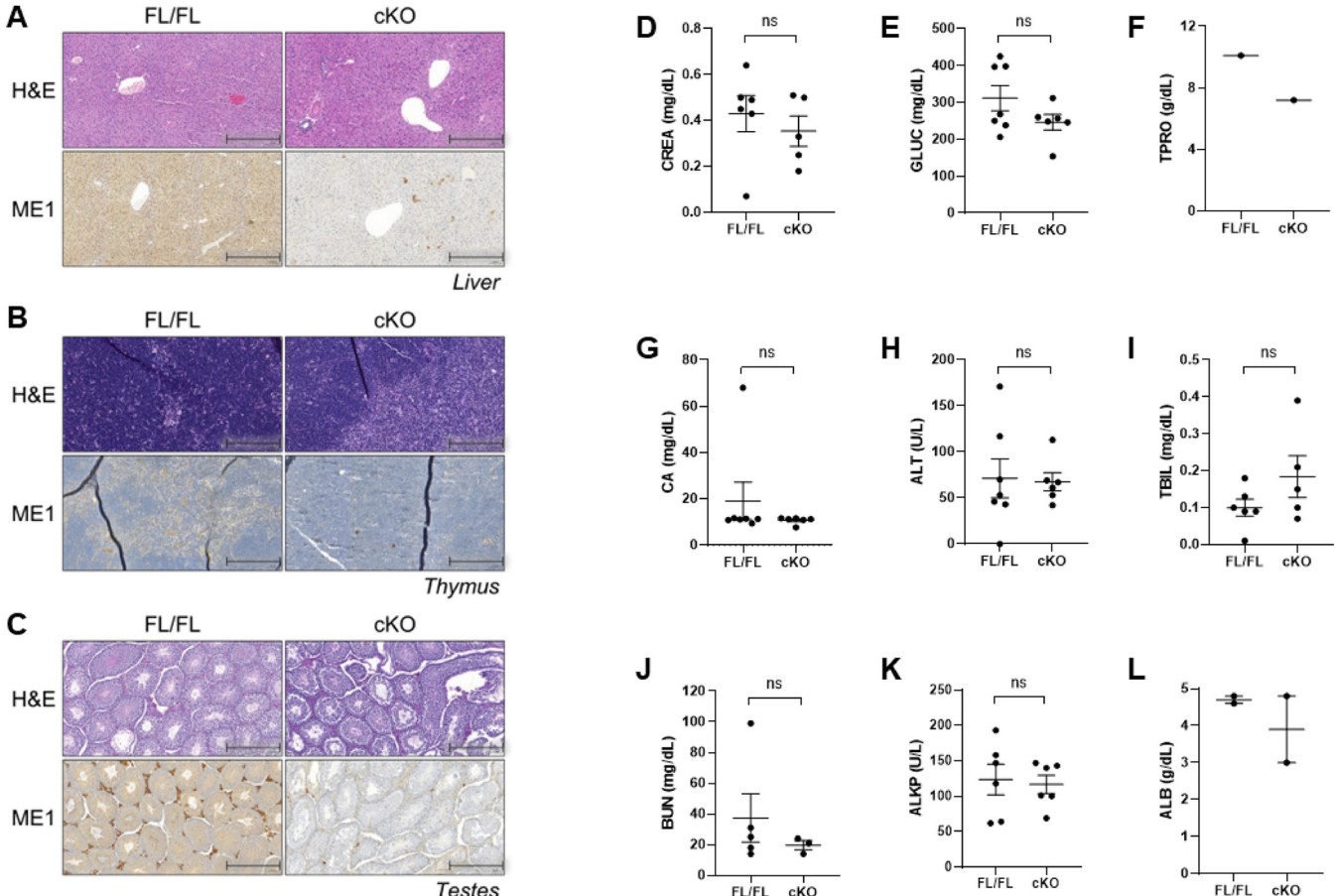

**Fig 5. ME1 protein expression is notably reduced in cKO mice with no hematological phenotype.** (A-C) Me1 FL/FL and Me1 cKO mice were examined for histology, immunohistochemistry, and serum chemistry. Slides were stained with H&E and a 1:500 ME1 antibody stain ratio for IHC. Representative slides are presented. (A) The liver was imaged at 10X magnification (scale bar = 400 μm), (B) the thymus at 20X (scale bar = 200 μm), and (C) the testes at 10X (scale bar = 400 μm). Across the three tissues, Me1 KO mice exhibited significant loss of ME1. (D-L) Serum chemistry assessed for (D) CREA, creatinine; (E) GLUC, glucose; (F) TPRO, total protein; (G) CA, calcium; (H) ALT, alanine transferase; (I) TBIL, bilirubin; (J) BUN, blood urea nitrogen; (K) ALKP, alkaline phosphatase; and (L) ALB, albumin. Collection of insufficient blood volume resulted in lower sample size for TPRO and ALB.

can be studied for embryonic and early-age studies with similar applications to Mod-1 mice. The *Me1* conditional knockout model brings a novel approach to explore the effects of ME1 knockout in adulthood, simulating a therapeutic window for potential ME1 inhibitors. Further, the Me1 flox gene in the conditional knockout model can be combined with different Cre recombinases to explore the effects of *Me1* knockout in specific tissues and organs. The Me1 KO and Me1 cKO models each contribute a replicable, genetically engineered knockout model with a similar level of efficacy and increased flexibility in application.

## Me1 germline knockout model phenotype

The Me1 KO model examined the effects of *Me1* knockout in the germline. Much like the Mod-1 mutant model, Me1 KO mice age to adulthood normally and reproduce for several generations without complications [20]. However, when heterozygous Me1 KO mice were bred together, a noticeable preference for transmission of the Me1 null allele was observed, indicating that a non-Mendelian transmission phenotype may be involved. Further examination revealed that the TRD takes place almost exclusively when transmitted maternally to male

progeny. This pattern is consistent with X-linked genes, but to our knowledge, no strong relation between *Me1* and the X chromosome has been observed. The TRD observed here is considerably less penetrant than the t complex phenotype, in which mutant transmission was selected for in up to 99% in some cases [29]. Further investigation into the mechanism of maternal influence in male progeny remain the focus of future studies.

## Me1 conditional knockout model phenotypes

The Me1 cKO model showed little to no adverse phenotype, opening the door for future studies to explore the effects of *Me1* knockout in adulthood. Malic enzyme 1 has been implicated to play a role in obesity [24], type 2 diabetes [47], and even cancer development [3, 7, 48]. This is largely due to its production of cytosolic NADPH, driving lipogenesis and managing redox balance [7]. Here, we strived to identify any subtle phenotypes of *Me1* knockout in adulthood.

The Me1 cKO model thoroughly demonstrated a lack of phenotype when *Me1* is knocked out in adulthood and aged under normal conditions. Through the PhenoMENA Clinical Examination and Behavioral Assessment, Me1 cKO mice showed appropriate behaviors similar to their *Me1*-expressing controls. Grip strengths were also statistically equivalent between groups. Non-invasive blood pressure measurements revealed similar circulatory abilities, additionally scoring within in a normal range for C57BL/6J mice [49]. Further, histological analysis of several Me1 cKO tissues were not distinct from their controls. Orthotopic injection of KPC cells did not reveal that Me1 cKO mice provide a more conducive environment for tumor growth than mice with the functional gene. The rotarod performance test revealed a slight difference between the Me1 cKO group and controls. One notable limitation to this test is acclimation time. In many behavioral studies, rodents are pre-trained on the accelerating rotarod apparatus prior to recorded trials, with more training time correlating to more consistent performance [50]. In this study, mice were not pretrained, so it was recommended that best times be recorded for each mouse rather than averages. This resulted in high standard deviations in both groups. Since the mean performances were well within one standard deviation of each other, it may stand to indicate that further test acclimation was required prior to testing. In future studies, a longer acclimation phase is recommended to mitigate any learning curve involved with using the apparatus.

## Limitations of normal conditions

The lack of an observed phenotype in Me1 deficient mice is in marked contrast to expectation for such an evolutionarily conserved gene. Malic enzymes are widely present in animals and plants, with some sequences even tracing back from humans to yeast [2, 6, 51, 52]. It would naturally follow that Me1 plays an essential role in organismal survival, and that its loss or mutation would necessarily be a detriment to carriers. Instead, we see that its loss is preferred in some cases. For that reason, it is important to note that these tests were performed on mice raised under standard laboratory conditions, devoid of many stressors in the natural world. Since survivability is a primary driver of evolutionary conservation [53], testing these models with stress-inducing trials encountered in nature, such as modified food availability or infection, could reveal the role of ME1 in vivo.

## Conclusion

In this paper, we describe two new mouse models to knockout malic enzyme 1 in the germline and adulthood. These models expand upon the current body of tools to explore the effects of ME1 in vivo. Application of *Me1* knockout mice in a large battery of molecular, histological, and behavioral tests revealed that *Me1* knockout did not have an observable impact under

standard laboratory conditions. Rather, *Me1* loss seems preferential to the functional gene when maternally transmitted in the male germline. This surprising observation stands in contrast to the evolutionary conservation of malic enzymes in prokaryotes and eukaryotes alike. Thus, the testing of these *Me1* knockout mice in stressful conditions, crossing alternative Cre recombinases into the Me1 flox framework, and exploring the apparent Me1 KO transmission ratio phenotype could shine further light on the critical roles of Me1 in vivo.

## Supporting information

**S1 Table. Me1 KO mice age to adulthood.** Me1 KO mice were aged to adulthood and were euthanized after the 100-week timepoint. Heterozygous mice for the Me1 null allele were similarly aged to adulthood, but some were euthanized before the 100-week timepoint due to necessary consolidation measures in 2020.
(DOCX)

**S2 Table. Raw data.** All raw data for figures, tables, and supplementary figures.
(XLSX)

**S1 Fig. Developmental body weight gain of Me1 KO mice on a standard laboratory diet.** Body weights of wild-type male (WT-M; n = 3) and female (WT-F; n = 3) mice were tracked Me1 KO male (KO-M; n = 7) and female (KO-F; n = 3) mice weekly from 3–8 weeks of age. An initial discrepancy in body weight at 3 weeks of age between the WT-M and KO-M mice was noted specifically in this cohort, but body weight gain and growth rates did not reach significant differences at any time point.
(TIF)

**S2 Fig.** NADP+, NADPH, and NADP+/NADPH ratios in (A) liver, (B) thymus, (C) testes, (D) kidney, (E) pancreas, and (F) spleen from Me1 wild type versus Me1 null animals, as assessed by LC-MS and presented by sex. Pooled samples were included as controls; blanks assessed instrument performance and represent background without biological sample.
(TIF)

**S3 Fig. Histology of tissues obtained from Me1 cKO mice show no pathological phenotype.** Kidneys imaged at 5X (scale bar = 400 μm). Stomach, cecum, and large intestine imaged at 20X (scale bar = 100 μm). All other tissues imaged at 10X (scale bar = 200 μm).
(TIF)

**S4 Fig. Raw, uncropped gels.** Uncropped gel images from Fig 1D–1F.
(PDF)

## Acknowledgments

We acknowledge Thomas L. Saunders, Elizabeth Hughes, Wanda Filipiak, Galina Gavrilina, Honglai Zhang and the Transgenic Animal Model Core of the University of Michigan's Biomedical Research Core Facilities for design and production of the Me1 chimeric mice. We would also like to thank the members of the In Vivo Animal Core in the Unit for Laboratory Animal Medicine at Michigan Medicine for their necropsy, hematology, histology, and pathology expertise.

## Author Contributions

**Conceptualization:** Jonathan M. Alektiar, Mengrou Shan, Lewis C. Cantley, Jacob L. Mueller, Costas A. Lyssiotis.

**Data curation:** Jonathan M. Alektiar, Mengrou Shan, Li Zhang, Christopher J. Halbrook, Ivan F. Mier.

**Formal analysis:** Jonathan M. Alektiar, Mengrou Shan, Li Zhang, Christopher J. Halbrook.

**Funding acquisition:** Costas A. Lyssiotis.

**Investigation:** Jonathan M. Alektiar, Mengrou Shan, Megan D. Radyk, Li Zhang, Lin Lin, Carlos Espinoza, Ivan F. Mier, Brooke L. Lavoie, Lucie Salvatore, Jacob L. Mueller, Costas A. Lyssiotis.

**Methodology:** Jonathan M. Alektiar, Megan D. Radyk, Carlos Espinoza, Ivan F. Mier, Lucie Salvatore, Jacob L. Mueller, Costas A. Lyssiotis.

**Project administration:** Megan D. Radyk, Marina Pasca di Magliano, Lewis C. Cantley, Jacob L. Mueller, Costas A. Lyssiotis.

**Resources:** Lewis C. Cantley, Jacob L. Mueller.

**Supervision:** Marina Pasca di Magliano, Lewis C. Cantley, Costas A. Lyssiotis.

**Writing – original draft:** Jonathan M. Alektiar, Costas A. Lyssiotis.

**Writing – review & editing:** Jonathan M. Alektiar, Mengrou Shan, Li Zhang, Christopher J. Halbrook, Lin Lin, Lucie Salvatore, Lewis C. Cantley, Jacob L. Mueller, Costas A. Lyssiotis.

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
