## [Decision Letter · Decision Letter 0]

13 Sep 2023

PONE-D-23-20509Malic enzyme 1 knockout has no deleterious phenotype and is favored in the male germline under standard laboratory conditionsPLOS ONE

Dear Dr. Lyssiotis,

Thank you for submitting your manuscript to PLOS ONE. After careful consideration, we feel that it has merit but does not fully meet PLOS ONE’s publication criteria as it currently stands. Therefore, we invite you to submit a revised version of the manuscript that addresses the points raised during the review process,  particularly the NADPH/NADP+ or triglyceride levels in tissues.

We look forward to receiving your revised manuscript.

Kind regards,

Junming Yue

Academic Editor

PLOS ONE

Journal Requirements:

"M.S. was supported by an F32 fellowship from the NIH/NCI (5F32CA247492). M.D.R. receives funding from the National Institute of Child Health and Human Development (Training Program in Organogenesis, T32HD007505) and the National Cancer Institute (F32CA275283). C.A.L and L.C.C. were supported by a Mark Foundation ASPIRE award and a research partnership alliance with Astellas Pharmaceuticals. 

We acknowledge Thomas L. Saunders, Elizabeth Hughes, Wanda Filipiak, Galina Gavrilina, Honglai Zhang and the Transgenic Animal Model Core of the University of Michigan’s Biomedical Research Core Facilities for design and production of the Me1 chimeric mice. Research reported in this publication was supported by the University of Michigan Transgenic Animal Model Core and the Biomedical Research Core Facilities and by the National Cancer Institute of the National Institutes of Health under award number P30CA046592. We would also like to thank the members of the In Vivo Animal Core in the Unit for Laboratory Animal Medicine at Michigan Medicine for their necropsy, hematology, histology, and pathology expertise. PhenoMENA, non-invasive blood pressure, and rotarod testing was performed by the Physiology Phenotyping Core at the University of Michigan Medical School, which is supported in part by the Michigan Musculoskeletal Health Center (NIH P30 AR069620). The funders had no role in study design, data collection and analysis, or the content and publication of this manuscript."

"M.S. was supported by an F32 fellowship from the NIH/NCI (5F32CA247492). M.D.R. receives funding from the National Institute of Child Health and Human Development (Training Program in Organogenesis, T32HD007505) and the National Cancer Institute (F32CA275283). C.A.L and L.C.C. were supported by a Mark Foundation ASPIRE award and a research partnership alliance with Astellas Pharmaceuticals. 

We acknowledge Thomas L. Saunders, Elizabeth Hughes, Wanda Filipiak, Galina Gavrilina, Honglai Zhang and the Transgenic Animal Model Core of the University of Michigan’s Biomedical Research Core Facilities for design and production of the Me1 chimeric mice. Research reported in this publication was supported by the University of Michigan Transgenic Animal Model Core and the Biomedical Research Core Facilities and by the National Cancer Institute of the National Institutes of Health under award number P30CA046592. We would also like to thank the members of the In Vivo Animal Core in the Unit for Laboratory Animal Medicine at Michigan Medicine for their necropsy, hematology, histology, and pathology expertise. PhenoMENA, non-invasive blood pressure, and rotarod testing was performed by the Physiology Phenotyping Core at the University of Michigan Medical School, which is supported in part by the Michigan Musculoskeletal Health Center (NIH P30 AR069620). The funders had no role in study design, data collection and analysis, or the content and publication of this manuscript."

"L.C.C. and C.A.L. are inventors on patents pertaining to Kras regulated metabolic pathways, redox control pathways in pancreatic cancer, and targeting GOT1 or ME1 as a therapeutic approach (US Patent No: 2015126580-A1, 05/07/2015; US Patent No: 20190136238, 05/09/2019; International Patent No: WO2013177426-A2, 04/23/2015). L.C.C. owns equity in, receives compensation from, and serves on the Scientific Advisory Boards of Faeth Therapeutics, Agios Pharmaceuticals, Volastra Therapeutics, and Larkspur Biosciences. L.C.C.’s laboratory has previously received financial support from Petra Pharmaceuticals. Agios Pharmaceuticals is identifying metabolic pathways of cancer cells and developing drugs to inhibit such enzymes to disrupt tumor cell growth and survival. C.A.L. has received consulting fees from Astellas Pharmaceuticals, Odyssey Therapeutics, and T-Knife Therapeutics."

Reviewers' comments:

Reviewer's Responses to Questions

**Comments to the Author**

1. Is the manuscript technically sound, and do the data support the conclusions?

Reviewer #1: Yes

Reviewer #2: Yes

2. Has the statistical analysis been performed appropriately and rigorously? 

Reviewer #1: Yes

Reviewer #2: Yes

3. Have the authors made all data underlying the findings in their manuscript fully available?

Reviewer #1: Yes

Reviewer #2: Yes

4. Is the manuscript presented in an intelligible fashion and written in standard English?

Reviewer #1: Yes

Reviewer #2: Yes

5. Review Comments to the Author

Reviewer #1: Manuscript Number: PONE-D-23-20509

In the manuscript by Alektiar JM and Shan M, titled “Malic enzyme 1 knockout has no deleterious phenotype and is favored in the male germline under standard laboratory conditions”, the authors describe the interrogation of loss of malic enzyme 1 (ME1) using in vivo genetic models. The authors demonstrate that loss of ME1, either constitutively deleted during animal development or deleted in adult animals, had no detrimental effects on animals. Interestingly, they show that the ME1 KO genotype is favored when maternally transmitted to male progeny. This is a clear and thorough study; only minor suggestions are provided to improve the study.

Minor Suggestions

1. The authors do not measure NADPH/NADP+ or triglyceride levels in the same tissues where they observe ME1 genetic/protein deletion. Measuring NADPH/NADP+ and triglycerides would significantly add to the conclusions, especially since the authors discuss how ME1 is reported to promote NADPH generation and triglyceride production. Regardless of the outcome, this data would be very informative. If these metrics don’t change, it could suggest an explanation for the dispensability of ME1. If these metrics do change, it could potentially suggest that these changes are not sufficient to impair animal development and homeostasis.

2. The authors could improve the clarity of Figure 1 by adding details to Fig. 1D, 1E, and 1F to show, along with the mouse strains, what the PCR experiment is amplifying. For example, having “Me1 null lacZ reporter” in 1D and “loxP” in Figure 1E. Alternatively, the authors could include the description of the primer pairs used in Fig. 1D-1F in the targeting strategy schematics in Fig. 1A-1C.

3. It is unclear why the loss of protein expression is described as a “knockdown” of ME1 throughout the manuscript instead of “knockout”, which might be more appropriate.

Reviewer #2: The present manuscript describes 2 mouse models with lack of ME1. These models are used to validate that ME1 is not essential for normal development or physiological function, at least in the absence of stresses like infection or starvation. The data are objectively presented and compelling. There is also the surprising and intriguing observation that eggs lacking ME1 are preferentially transmitted, relative to WT, at least in the case of male offspring. This could possibly relate to an alternative NADPH pathway, G6PD, being X-linked, although why this would favor ME1 loss is unclear. Overall, this is a clear and valuable contribution and the mouse models will be valuable going forward for the community.

6. PLOS authors have the option to publish the peer review history of their article (what does this mean?). If published, this will include your full peer review and any attached files.

Reviewer #1: **Yes: **Isaac S. Harris

Reviewer #2: **Yes: **Josh Rabinowitz

---

## [Author Response · Author response to Decision Letter 0]

18 Apr 2024

Detailed responses to the referee comments are provided in the "Response to Reviewers" file appended to this revised submission.

---

## [Editor Report · Decision Letter 1]

29 Apr 2024

Malic enzyme 1 knockout has no deleterious phenotype and is favored in the male germline under standard laboratory conditions

PONE-D-23-20509R1

Dear Dr. Lyssiotis,

We’re pleased to inform you that your manuscript has been judged scientifically suitable for publication and will be formally accepted for publication once it meets all outstanding technical requirements.

Kind regards,

Junming Yue

Academic Editor

PLOS ONE
---

## [Editor Report · Acceptance letter]

29 May 2024

PONE-D-23-20509R1 

PLOS ONE

Dear Dr. Lyssiotis, 

I'm pleased to inform you that your manuscript has been deemed suitable for publication in PLOS ONE. Congratulations! Your manuscript is now being handed over to our production team.

Kind regards, 

on behalf of

Dr. Junming Yue 

Academic Editor

PLOS ONE